# A Systematic Review of MicroRNA Signatures Associated with the Progression of Leukoplakia with and without Epithelial Dysplasia

**DOI:** 10.3390/biom11121879

**Published:** 2021-12-14

**Authors:** Nadia Kaunein, Rishi Sanjay Ramani, Kendrick Koo, Caroline Moore, Antonio Celentano, Michael McCullough, Tami Yap

**Affiliations:** 1Melbourne Dental School, The University of Melbourne, 720 Swanston Street, Carlton, VIC 3053, Australia; rramani@student.unimelb.edu.au (R.S.R.); kendrick.koo@unimelb.edu.au (K.K.); moore@unimelb.edu.au (C.M.); antonio.celentano@unimelb.edu.au (A.C.); m.mccullough@unimelb.edu.au (M.M.); 2Royal Melbourne Hospital, Victorian Comprehensive Cancer Centre, Parkville, VIC 3053, Australia

**Keywords:** biomarkers, dysplasia, leukoplakia, prognosis, microRNA, oral potentially malignant disorders, oral cancer, malignant transformation

## Abstract

Oral cancer is a significant public health issue, being the eighth most common cancer worldwide with over 300,000 cases diagnosed annually. Early diagnosis and adequate management of oral potentially malignant disorders (OPMDs) before transformation into oral squamous cell carcinoma (OSCC) is critical to reduce deaths, morbidity, and to improve overall prognosis. MicroRNAs (miRNAs) are small noncoding RNAs involved in the post-transcriptional regulation of protein expression and implicated in the control of numerous cellular pathways and impacting physiological, developmental, and pathological processes. Dysregulation of miRNAs has been reported in many cancers and has been demonstrated to play a critical role in cancer initiation, progression, apoptosis, invasion and metastasis. This systematic review provides a comprehensive summary of the prevailing literature on miRNA signatures in OPMDs, specifically leukoplakia with or without oral epithelial dysplasia, and their utility in predicting malignant transformation into OSCC. Eighteen articles describing 73 unique and differentially expressed microRNAs met the criteria for inclusion in this review. We reviewed the characteristics and methodology for each of these studies and assessed the sensitivity and specificity of the studied miRNAs in predicting malignant transformation. This systematic review highlights the significant interest in miRNAs and their tremendous potential as prognostic markers for predicting the malignant transformation of OPMDs into OSCC.

## 1. Introduction

Oral squamous cell carcinoma (OSCC) is by far the most common histological subtype of oral cancer, accounting for 90–95% of all cases, and is the eighth most common cancer globally, with an annual incidence of over 300,000 cases [1,2]. Despite steady improvements in treatment, OSCC is still associated with a poor prognosis due to both local aggressiveness and metastatic progression [3]. Detection of OSCC at an early stage is the most effective strategy to reduce both mortality and morbidity of this disease [4].

A significant number of OSCCs develop from a group of precursor lesions [5] termed “potentially malignant disorders” by the World Health Organization (WHO) and encompassing lesions or conditions at risk of malignant transformation [6]. The most common of these oral potentially malignant disorders (OPMD) are leukoplakia, oral lichen planus, and actinic cheilitis [7]. Oral leukoplakia (OLK) presents as a white area or plaque that cannot be characterized as any other lesion clinically or histologically, and is the most common OPMD, with a prevalence between 0.6% to 4.6% [8]. A recent review suggested that the malignant transformation rate for OLK in total population ranges between 0.13% and 34.0% [9]. The current standard for assessing OLK and other OPMDs is the microscopic examination of intraoral biopsies and identification of architectural and cytological changes, referred to as oral epithelial dysplasia (OED) [10]. OED can be graded as mild, moderate, severe or carcinoma in situ based on these architectural and cytological changes [11]. Understanding the molecular mechanisms underpinning the progression into OSCC through these four stages is critical for an early and timely diagnosis. Although OLK and OED may progress to OSCC, clinical and histological examination have limited prognostic value and cannot reliably predict which lesions will progress [12]. The early diagnosis and adequate management of OPMD prior to progression to OSCC is critical for reducing mortality and morbidity and to improve overall prognosis [4,13].

MicroRNAs (miRNAs) are small, noncoding RNAs that are 18–25 nucleotides long and regulate protein expression at the post-transcriptional level [14]. Interaction between miRNAs and mRNAs can regulate the expression of multiple proteins or target genes, while concurrently the expression of a single target gene can be regulated by multiple miRNAs [15]. It is estimated that about 50% of all human genes are regulated by miRNAs, which thereby effect control upon numerous cellular pathways and physiological, developmental, and pathological processes [16]. Interaction of miRNA with a messenger RNA transcript can result in both gene silencing and gene activation. The role of miRNAs in oncogenesis has received significant attention in the last decade [17] and they can be broadly classified as “tumor suppressor miRNAs” or “onco-miRs” depending on their target genes [18]. Interestingly, a number of miRNAs have a dual role as both a tumor suppressor miRNA and onco-miR, depending on the tumor type and cellular context [19]. Dysregulation of miRNAs has been investigated in a range of cancers and has been found to play a critical role in cancer initiation, progression, apoptosis, invasion and metastasis [20]. In OSCC, miRNAs have been implicated in tumorigenesis with the identification of distinct miRNA expression profiles [21,22,23].

The diagnostic value of dysregulated miRNA signatures in OSCC points to the possible value of aberrant miRNAs as prognostic markers for malignant transformation in OPMDs [24]. Recent studies have demonstrated differentially expressed miRNAs in oral epithelial dysplasia and other OPMDs that predict the risk of recurrence and malignant transformation [13,25]. There has been increasing interest and investigation into miRNAs as potential prognostic biomarkers for OPMD over the last decade, providing impetus for the current work. This paper systematically reviews the prevailing literature of miRNA signatures in OLK and OED, and their value in predicting the malignant transformation of OPMDs to OSCC.

## 2. Materials and Methods

This systematic review was conducted and reported according to the Preferred Reporting Items for Systematic Reviews and Meta-Analyses (PRISMA) guidelines [26]. Electronic literature searches were conducted in Medline (Ovid), Embase (Ovid), Evidence Based Medicine (EBM) Reviews (Ovid), and Web of Science databases on 14 October 2020 with no restrictions placed on date of publication (Search strategies detailed in Table 1). The identified citations were imported into a reference management software package (Endnote X8, Clarivate Analytics, Philadelphia, PA, USA) and duplicate entries removed. Screening was conducted independently by two reviewers (NK, RR) who were blinded to each other. The titles of 526 articles were screened to exclude ineligible studies and 116 abstracts were reviewed to exclude studies outside the scope of this review. Full text review was conducted for 32 included studies. At each step, any inter-reviewer disagreements were resolved by a third reviewer (TY). Data was extracted from 18 included studies using a standardized data collection template.

### Selection Criteria

Studies were included if they discussed OLK or OED, alone or as a subset of OPMDs, and the prognostic value of miRNAs to predict the transformation of these oral lesions to OSCC. Study designs included those that compared miRNAs expression cross-sectionally in OLK/OED to OSCC or progressive versus non-progressive OLK/OED, as well as longitudinally in progressive OLK/OED with the subsequent paired OSCC. Findings from in vitro and animal studies were also eligible for inclusion. Exclusion criteria included non-English language articles, unpublished articles, withdrawn/retracted studies, reviews, case reports, conference abstracts, commentaries, opinion articles, and letters to the editor. Articles that included non-OLK or non-OED OPMDs but without available sub-analyses for these groups were excluded.

Data extracted from eligible studies included study characteristics, methodologies implemented, key miRNA findings, and data on miRNAs predicting malignant transformation in OLK or OED. The following data pertaining to the study characteristics were extracted: first author, year of publication, type of OPMD investigated (OLK or OED), specimen type, oral anatomical sub-site, grade of dysplasia, sample size, sample breakdown, follow-up time, presence of normal controls, any statistical analysis between high-risk vs. low risk sites, paired OLK/OED/OSCC cases with non-cancerous mucosa in the same patient, and paired OLK/OED and subsequent OSCC samples (Table 2).

Data pertaining to methodology and miRNA findings included the following categories: miRNAs investigated, source of miRNAs, miRNA discovery phase method, miRNA validation method, housekeeping control to normalize the miRNA expression values, direction of miRNA dysregulation, fold change between normal vs. OLK/OED, fold change between OLK/OED vs. OSCC, fold change between normal vs. OSCC, comparison between progressive and non-progressive OLK/OED (Table 3 and Table 4). With respect to miRNAs predicting malignant transformation in OLK and OED, the following data were extracted: sensitivity, specificity, receiver operating characteristic (ROC), and area under curve (AUC) (Table 5).

## 3. Results and Discussion

A total of 18 eligible studies published between 2008 and 2018 were identified and included in this review, describing a total of 1779 samples (OLK/OED, OSCC, normal) (Table 2 and Table 3). The sample size for the included studies ranged from 10 to 226 when control subjects were included, with 11 of the 18 studies having samples sizes greater than 70. The number of cases in each study ranged from 5 to 80 for OLK/OED and 3 to 118 for OSCC/HNSCC.

A range of techniques were used by these 18 studies in the discovery phase for the selection of miRNAs for investigation (Table 3). Literature search was the most common method, followed by TaqMan Low Density Array (TLDA), database search, global profiling, microarray analysis, small and total RNA next generation sequencing (NGS), among others. One study did not specify the method used in the discovery phase [27]. Six studies examined miRNA expression in serum, 13 studies considered tissue samples, and four studies analyzed saliva samples. Five studies investigated miRNAs in more than one biological source material [13,28,29,30,31]. Of the 17 studies that identified miRNA biomarkers with predictive or prognostic potential for malignant transformation in OLK/OED, only five of these studies explored biomarker sensitivity and specificity between different tissues involved in oral carcinogenesis (Table 5).

The selection and inclusion of appropriate reference miRNAs within miRNA panels for normalization has a significant impact on downstream analysis of miRNA expression [32]. A wide range of housekeeping controls have been utilized for normalization of miRNA expression data in the included studies, with RNU-44 being the most common (6 out of 18 studies). Three studies implemented U6, while RNU6B and RNU48 were used in two others. Other housekeeping controls included were miR-130b-3p, miR-221-3p, miR19b, miR31, miR205, miR-210, miR-16, SNORD68, RNU6, let-7a, and RNU19 were each utilized in a single study only. Pre-analytical and analytical variables such as haemolysis, spiked or internal-control miRNAs to name a few, can further influence the analysis of miRNAs [33]. MiR-16, for instance, which has been utilized by Hung et al., is highly susceptible to haemolysis and has been shown to not be a reliable internal control [33]. Further, Chen et al. and Kao et al. utilized U6, which has subsequently been abandoned due to little stability and susceptibility to degradation [33]. Similarly, another common control, RNU6B, has been observed to be unstable [33]. Conversely, three studies (Prasad et al., Chang et al. and De Sarkar et al.) used more than one housekeeping control in their experiments, which reportedly increases the efficiency of normalization when compared to using a single control [33].

### 3.1. Differentially Expressed miRNAs

A total of 73 unique differentially expressed miRNAs were reported by the 18 studies, with miR-21 and miR-31 being the most frequently examined miRNAs found to be differentially expressed. Fifteen of these differentially expressed miRNAs (20.5%) were reported in at least two studies (Table 4), with the remaining 58 reported in only a single study each (Appendix A). Of the 15 miRNAs investigated in at least two papers, nine miRNAs (60%) were dysregulated in a consistent direction with seven upregulated and two downregulated. The direction of change in expression was inconsistent for six miRNAs, or the expression was stage dependent (e.g., cancer progression stage, grade of dysplasia) (Table 4).

### 3.2. Quality of the Studies/Methods

Paired OPMD and OSCC samples from the same participants would be able to account for inter-patient variations and a host of other confounding factors and therefore ideal for the investigation of miRNAs with prognostic value in predicting malignant transformation with increased statistical significance and improved study design. However, such samples may be harder to access. Only one third of the included studies had paired OLK/OED and OSCC samples [34]. Yang et al., for example, used paired OLK and OSCC samples and reported upregulation of 12 miRNAs and downregulation of 13 miRNAs in progressive lesions compared to non-progressive lesions [28]. Similarly, Harrandah et al. used progressive OPMD with paired sequential OSCC tumours from the same site and concluded that miR-375 downregulation was significantly associated with malignant transformation in OPMDs. Further, Harrandah et al. excluded cases if the precursor lesion and OSCC location were not the same, thereby eliminating the potential confounding factor of oral subsite [35].

Defining ideal controls for studies investigating malignant transformation is far from straightforward. In addition to the ethical challenge in obtaining a biopsy for normal healthy tissue, there is limited information on the miRNA profiles of other diseases that can further limit comparison between pathologies. Only one study, Zahran et al. (2015), included mucosal disease controls (recurrent aphthous stomatitis) in the sample in addition to normal controls [36]. Two studies (Chang et al. 2008, De Sarkar et al., 2014) included a paired non-cancerous mucosa from adjacent clinically normal site [37,38]. De Sarker et al. included both normal controls and paired adjacent non-cancerous mucosa in their study, while Chang et al. included only paired adjacent clinically normal mucosa as controls [37,38].

Half the papers reviewed (9/18) were cross-sectional studies for which follow-up time was not applicable [27,30,38,39,40,41,42,43,44] and three studies [25,35,37] did not include control subjects in their study design. Whilst cross-sectional observational studies allow for larger sample sizes and obviate the need for follow up, longitudinal follow up of OPMD patients to record recurrence or malignant transformation would allow a more robust appraisal of the potential of miRNAs as prognostic markers.

Numerous other non-OSCC, non-OPMD related factors can influence miRNA dysregulation, and recent studies have linked dysregulation of numerous miRNAs to other oral diseases such as periodontitis, inflammatory diseases and Sjogren’s syndrome [45]. Additionally, certain medications and immunotherapy have also been demonstrated to influence miRNA expression [46]. These factors were not considered or adjusted for in data analysis in any of included studies.

A further methodological issue was the heterogenous sampling and concatenation of different OPMDs into a single group. Although all studies reported the different OPMDs included in the sample, subgroup analysis investigating differences between different OPMDs was either not considered or could not be performed due to small numbers. This is particularly critical as it has previously been reported that miRNA dysregulation can act as both onco-miR and tumor suppressor depending on stage of disease [19]. However, 11 studies specified the grade of dysplasia or type of OLK included in their samples. None of the included studies considered a sub-group analysis to investigate the differences in miRNA signatures in different types of OLK. Of the 11 studies that investigated dysplasia, six studies [13,25,35,36,38,47] included OPMDs or OLKs that were histopathologically categorized as dysplastic and non-dysplastic in their study sample. Five studies [13,25,29,36,47] compared miRNA expression between dysplastic versus non-dysplastic lesions. Zahran et al., for instance, reported no statistical difference in relation to miR-145 between dysplastic and non-dysplastic OPMDs [36]. However, when the same two groups were compared regarding miRNA-21 and miRNA-184, the difference was statistically different for both miRNA’s (*p* < 0.05) [36]. On the other hand, Hung et al. inferred that miR-31 was up-regulated independently from development of epithelial dysplasia [13].

The remaining four studies, 28–31, included only dysplasia cases in their study sample. A total of five studies [29,31,36,42,47] assessed the association between severity/grades of dysplasia and miRNA expression. Cervigne et al. deduced that there was a consistent increase in miRNA-21, miR-181b, miR-345 expression associated with severity of dysplasia [47]. Similarly, Kao et al. reported on expression of microRNAs in dysplasia, specifically that miR-146a, miR-184, and miR-372 “were detected in early stages of transformation and significantly eminent at the most advanced lesion state” [29]. Presumably this would indicate that these microRNAs were present in mild dysplasia with increasing levels in higher levels of dysplasia and OSCC. Interestingly, Brito et al., undertook a detailed investigation correlating histological changes with microRNA changes, mapping increased expression of miR-21, miR-181b, and miR-345 with specific pathological features of dysplasia [31]. Nevertheless, they found that this was not inclusive, and that not all histological parameters are associated with significant molecular alterations [31]. Therefore, due to the overlap of sampling methodology in the included studies, OLK and dysplasia are often grouped together, and have been grouped for this review. There is logic in preferring a biomarker that provides risk-stratification of OLK independent of dysplasia with the potential of further being combined with histopathological findings for prediction.

All studies bar one assessed miRNA expression in OPMDs and OSCC. The study that did not include OSCC samples (Philipone et al.), retrospectively analyzed miRNAs in OLK lesions that were classified into transformed or non-transformed [25]. For a specific miRNA to be utilized as a good prognostic biomarker, it should differentiate between progressive and non-progressive precursor lesions by exhibiting differential expression. Therefore, despite not having analyzed OSCC samples, this study had a robust study design to investigate lesions at risk of cancer progression.

There have been conflicting reports on the prognosis of OPMDs and risks of malignant transformation into OSCC for lesions in different oral subsites, such as floor of the mouth and the oral tongue. A few studies have attributed factors specific to the location of the lesion to the risk of malignant transformation. While some authors have reported a higher risk of malignant transformation of OLK in the floor of the mouth and tongue, others found no oral subsites had a higher risk [48]. Only 11 authors in the present review reported the location of OLK/OED or OSCC. However, only three of these 11 studies included any analysis or discussion between high risk versus low-risk sites. Harrandah et al. found that a lower risk of malignant transformation was seen when lesions were located in the buccal mucosa, gingiva, vestibule, palate, or dorsal tongue, while a higher risk was associated with floor of the mouth, ventral tongue or lateral tongue [35]. Similarly, Philipone et al. reported that 65% of all progressive lesions were located in high-risk sites, the tongue and floor of mouth [25]. Although these papers [25,40,47] commented on the homogeneous and non-homogeneous type of OLK in their respective samples, none of the included papers performed a sub-group analysis to investigate differences in miRNA signatures according to clinical appearance. It would be interesting to investigate the association between miRNA dysregulation, oral subsites, and malignant transformation risk in future studies.

Comparisons of miRNA signatures in progressive or non-progressive OLK/OED either prospectively or retrospectively is required to make definitive conclusions about miRNA dysregulation and risk of malignant transformation. Only studies that directly or indirectly assessed this were included in this review, although this was not the primary objective of some of the included studies. Nevertheless, six out of 18 studies compared miRNA signatures between progressive and non-progressive lesions. Chattopadhyay et al. observed that miR-31 expression was higher in progressive OLK group when compared to the non-progressive OLK group [27]. A similar conclusion was reached by Hung et al. who reported an upregulation of miR-31 levels predicting OPMD progression. However, this association was only evident in patients that had epithelial dysplasia and not in patients without dysplasia. The authors suggest that miR-31 and epithelial dysplasia could synergistically predict OPMD progression [13]. Furthermore, not all studies specified exclusion of subjects with current or previous malignancy and previous resection of OLK/OED, which was a clear limitation.

Apart from the miRNA specific issues raised above, there are several more general methodological concerns. The small sample size in some of the studies could introduce selection bias and threaten the validity and generalizability of the studies’ results [49]. Additionally, the recording of all known confounding variables is critical to confirm a definite association or causality. Confounders such as age, sex, subsites, smoking status, and other oral/general health conditions need to be recorded so as to necessarily adjust for these factors while analyzing the data [50], and these were not uniformly included in all studies reviewed.

### 3.3. Extra-Oral miRNA

Although microRNAs exist stably in various bodily fluids, systematic differences between sample types are largely unknown [33]. There have been contradictory reports pertaining to the concentration of miRNAs in biofluids such as serum and plasma, for instance. Previously, it was reported that there is no difference in miRNA concentration between plasma and serum samples, however, recent reports suggest that miRNA concentration is higher in plasma when compared to serum [33]. These differences could possibly result from phlebotomy or sample processing protocols and highlight the need for a detailed sub-group analysis of sample type. The majority of the included studies in our review sourced miRNAs from oral biopsy tissue (13 out of 18), five studies utilized blood samples, serum or plasma, and four studies analyzed saliva. Five of the 18 studies used more than one of the above miRNA sources, and interestingly none of these five studies commented on any significant differences in miRNA signatures based on sample type. While some authors used different sample types for the discovery and validation phase, one study, Hung et al., reported the expression of miRNAs across different sample types, demonstrating that miR-21 and miR-31 showed consistent increased expression across tissue and saliva samples [13].

### 3.4. MiRNAs and Prediction of Malignant Transformation of OLK/OED

MiRNA dysregulation in OLK/OED was reported in all included studies, with promising miRNA signatures that can predict the risk of malignant transformation. However, only five studies investigated the sensitivity, specificity, and AUC of miRNAs in predicting malignant transformation in OLK/OED [13,25,35,40,44]. Four other studies included such analyses to validate the diagnostic power of miRNAs and were therefore excluded from Table 5 as the focus was not solely on ‘prognosis’ but rather ‘diagnosis’ of OLK/OED.

Including multiple biomarkers in a model improves the prognostic capability of miRNA expression over a single biomarker. One of the most investigated miRNAs, miR-21, demonstrated a sensitivity of just 51.61% and AUC of 0.651 in predicting malignant transformation of OLK/OED in tissue samples [35]. The prognostic value of miR-21 is seemingly further diminished when considering Hung et al., that demonstrated miR-21 to be statistically non-significant as an independent biomarker comparing progressive and nonprogressive OPMDs [13]. However, Harrandah et al. analyzed the effect of miR-21 in combination with miR-375, yielding a sensitivity of 75.76%, specificity of 100% and AUC of 0.925 for predicting malignant transformation [35]. Similarly, Lu et al. studied a combination of miR-196a, miR-196b alone as well as a combination of both miR-196a and miR-196b (together) to analyze the effectiveness of these miRNAs as prognostic biomarkers. The study could clearly segregate miR-196a and miR-196b expression levels between healthy vs. OPMD group and healthy vs. progressive cancer group. In addition, the combined determination of miR-196a and miR-196b demonstrated excellent potential to predict malignancy with sensitivity of 91%, specificity of 85% and AUC = 0.950, when compared to using miR-196a alone, which had low sensitivity of 64.2%. While miR-196a served as an excellent biomarker for detecting specificity (96.2%), miR196b had high sensitivity (93.4%) and the two molecules therefore complemented each other, improving their predictive power [40]. A further study using a panel of miRNAs was a retrospective analysis by Philipone et al. [25]. The samples were analyzed retrospectively after a follow up period of five years, and a combination of miRNAs (miR-208b-3p, miR-3065-5p, miR-129-2-3p, miR-204-5p) in a panel, while adjusting for age and histologic diagnosis, to identify non- and low-grade dysplastic lesions at risk of cancer progression. The predictive value for AUC was 0.792, with sensitivity and specificity of 76.9% and 73.7%. Another panel of miRNAs involving miR-129-5p, miR-296-5p and miR-450b-5p used by Chen et al. yielded individual AUC values of 0.730, 0.759 and 0.721, respectively, to discriminate OLK lesions from OLK-OSCC transformed lesions [44]. Interestingly, when all three miRNA signatures were combined, this panel demonstrated high accuracy with an AUC value of 0.872 (Table 5) [44]. However, further validation is warranted with a larger cohort. These findings are in line with previous reports that recommend using a combination of miRNAs or multiple miRNAs in a panel in order to improve prognostic ability of miRNAs [51,52]. No studies utilized the same combined panel of miRNAs. In contrast, certain miRNAs such as miR-375 could independently and clearly differentiate between progressive and non-progressive premalignant lesions with a *p* value of <0.0001, demonstrating excellent prognostic ability with 90% sensitivity, 100% specificity and 0.957 AUC [35]. Further, the authors suggested that miR-375 in premalignant lesions was a superior and useful predictor of the outcome, whereas pathologic assessment was not found to be useful in this study. Likewise, another independent biomarker with promising result is miR-31, which has been demonstrated to predict OPMD progression and malignant transformation with a sensitivity and specificity of 87.51% and 73.73% [13]. Consistently, ROC analysis revealed that miR-31 had an AUC of 0.81 (95% CI: 0.65 to 0.97), supporting the effectiveness of this biomarker in differentiating patients with progressive disease from non-progressive patients [13].

## 4. Future Directions and Conclusions

The present review has identified a substantial body of work suggesting the tremendous potential for miRNAs to be utilized as prognostic markers to predict the malignant transformation of OPMDs to OSCC. However, the quality of most of the included studies has been impacted by one or more limitations to the study design, such as small sample size, no subgroup analysis, heterogenous sampling and design, no use of diseased controls, use of unreliable internal controls, not excluding subjects with current or previous malignancy as well as previous resection of OLK/OED, and not adjusting for known confounding factors during data analysis. It is clear that single-center studies are unable to have a larger sample size that could address some of the aforementioned limitations, due to the low rate of malignant transformation. Moreover, the molecular data were not made openly available by all authors which has prevented us from conducting a formal meta-analysis. To circumvent these problems, Villa et al., suggested that a multi-level international collaborative research group that strategically designs, collects, shares data is necessary to achieve a larger sample size and therefore an adequately powered study population with a robust study design [53].

Although some miRNAs can independently differentiate between progressive and non-progressive lesions, this review has highlighted the importance of using a panel of multiple miRNAs that can complement each other’s sensitivity and specificity to significantly improve the prognostic ability. In addition, this review has identified and summarized a total of 73 unique and differentially expressed miRNAs that have been so far investigated for diagnosing OPMDs and predicting OSCC, with 15 miRNAs being reported in at least two studies and nine miRNAs shown consistent dysregulation in all study when they were investigated. Although all 73 uniquely dysregulated miRNAs (Table 4 & Appendix A) need further investigation, the nine miRNAs consistently across all studies (Table 4) warrant immediate attention for future studies. Furthermore, there is a need for more longitudinal studies to establish temporality in the relationship between miRNAs dysregulation and the malignant transformation of OLK/OED.

## Figures and Tables

**Table 1 biomolecules-11-01879-t001:** Search strategies applied to the databases: Medline (Ovid), Embase (Ovid), EBM Reviews (Ovid), Search strategy applied to the database: Web of science.

Search #	Query
1	(miRNA or microRNA or miR*).mp.
2	(oral or mouth or tongue or floor or palate or lingual or buccal or lip or labial or mucosa* or retromolar or cheek or gingiva or intra-oral or vermillion border).mp.
3	(cancer or neoplasm or squamous cell carcinoma).
4	(leukoplakia or white patches or erythroplakia or red patches or erythroleukoplakia or precancer* or epithelial dysplasia or OPMD or oral potentially malignant disorder or oral potentially malignant lesion or proliferative verrucous leukoplakia or pre-cancer* or pre-malignan* or dysplasia or premalignant).mp.
5	1 and 2 and 3 and 4
TOPIC: (miRNA OR microRNA OR miR*) AND TOPIC: (oral OR mouth OR pharyn* OR oropharyn* OR throat* tongue OR floor OR palate OR lingual OR buccal OR lip OR labial OR tonsil* OR mucosa* OR retromolar OR cheek OR gingiva OR intra-oral OR “vermillion border”) AND TOPIC: (Leukoplakia OR white patches OR erythroplakia OR red patches OR erythroleukoplakia OR precancer* OR epithelial dysplasia OR oral potentially malignant disorder OR oral potentially malignant lesion OR proliferative verrucous leukoplakia OR pre-cancer* OR pre-malignan* OR dysplasia OR premalignant OR pre-malignant*).	

**Table 2 biomolecules-11-01879-t002:** Study characteristics and key features of included studies.

Author (Year)	Disease Studied	Sample Size	Subject Break-Down	Normal Control (Yes/No)	Paired with Non-Cancerous Mucosa in Same Patient	Follow Up Time (Months)	Investigated Association between miR Expression and Severity of Dysplasia	Paired OLK/OED and OSCC	Oral Anatomical Sub-Site	Comparison between High Risk and Low Risk Sites
Studies comparing progressive vs non-progressive OLK/OED										
Cervigne et al. (2009)	OLK, OED	50	OLK-29, OSCC-14, Normal-7	Yes	No	60–108	Yes	Yes	Tonsil, Alveolus, Lip, FOM, Tongue, Buccal mucosa	No
Yang et al. (2013)	OLK, OED	52	OLK: 35, CIS/OSCC: 10, Controls: 7	Yes	No	36 – 60	No	Yes	Tongue, Palate, Buccal Mucosa, Lip	No
Harrandah et al. (2016)	OPMD/OED	37	OSCC: 6, OPMD-31, Controls (OPMD that did not progress to OSCC): 6	No	No	60	NS (Not specified)	Yes	Buccal mucosa, Tuberosity, Palate, Vestibule, Gingiva, Alveolar ridge, FOM, Tongue.	Yes
Chattopadhyay et al. (2016)	OPMD /OLK	61 (relevant to this topic)	OSCC: 23, OLK: 18, Normal-20.	Yes	No (compared results with previous study that had this data)	No	No	Unclear	NS	No
Hung et al. (2016)	OPMD/OED	93	(16 OED, 30 non-OED), OPMD Saliva: 20, OSCC: 3, Control (Saliva): 24	Yes	No	27.3	No	Yes (NA)	Buccal mucosa, gingiva, lip, palate, tongue	No (Briefly mentioned in discussion)
Philipone et al. (2016)	OLK, OED	77	OLK: 80, Control: 0	No	No	60	No	NA	Tongue, Floor of mouth, Buccal mucosa, Vestibule, Gingiva, Palate, Lip mucosa	Yes
Studies comparing OLK/OED with OSCC										
Chang et al. (2008)	OPMD	45	OSCC-39, OPMD-9 (4 OED, 5 EHP)	No	Yes	12.5 m for OSCC	No	No	Buccal Mucosa, Gingiva, Palate, tongue	No
Santhi et al. (2013)	OLK	164	OLK: 49, OSCC: 84, Normal: 31	Yes	No	No	No	No	NS	No
Brito et al. (2014)	OLK, OED	45	OLK: 22, OSCC: 17, Normal: 6	Yes	No	12	Yes	NS	Buccal Mucosa, Tongue, FOM, Soft Palate, Retro-molar trigone	No
DeSarkar et al. (2014)	OLK, OED	96	OLK: 18, OED-1, OSCC: 18, Other: OLP-11, Normal: 48	Yes	Yes	No	NS	NS	Gingiva, Buccal mucosa, Commissures	No
Kao et al. (2015)	OED	10 mice	5-Cases, 5-control	Yes	No	3.5	Yes	Yes	Tongue	No
Zahran et al. (2015)	OPMD/OED	100	OPMD: 40 (20-oed, 20: non-OED), OSCC: 20, Disease controls: 20, Normal controls: 20	Yes	No	36	No	No	Tongue, FOM, Alveolar margin, Retro-molar, Buccal mucosa.	No
Lu et al. (2015)	OPMD	159	OSCC: 90, OPMD: 16, Normal: 53	Yes	No	NS	No	No	NS	NS
Sun et al. (2016)	OLK	174	OSCC: 104, OLK: 30, Controls: 40	Yes	No	No	No	NS	For OSCC only: Tongue, non-tongue areas	Yes
Prasad et al. (2017)	OPMD/OED	70	OPMD: 30 (20: OED), OSCC: 20, Controls: 20	Yes	NS	No	Yes	No	NS	No
Chang et al. (2018)	OLK	178	OSCC: 82, OLK: 46, Normal: 50	Yes	No	No	No	No	No	No
Chen et al. (2018)	OLK	70	OLK: 30, OLK transformed OSCC: 25, Controls 15	Yes	No	NS	NS	Yes	NS	No
Wang et al. (2018)	OED	226	HNSCC: 118, OED: 48, Normal: 60	Yes	No	NS	No	No	NS	No

Abbreviations: OLK: Oral Leukoplakia; OED: Oral Epithelial Dysplasia; OSCC: Oral Squamous Cell Carcinoma; FOM: Floor Of the Mouth; CIS: Carcinoma-In-Situ; OPMD: Oral Potentially Malignant Disease; NS: Not Specified; NA: Not Applicable; EHP: Epithelial Hyperplasia; HNSCC: Head and Neck Squamous Cell Carcinoma.

**Table 3 biomolecules-11-01879-t003:** Methodology used in the included studies.

Author (Year)	MiRNA Investigated	Source (Serum /Plasma/Tissue/Saliva)	Method Performed in the Discovery Phase	Method Performed in the Validation Phase	Housekeeping Control
Chang et al. (2008)	miR-211, miR-204	Fresh frozen tissue	Literature	qRT-PCR TaqMan MicroRNA Assay (Applied Biosystems)	Let-7a miRNA, RNU19
Cervigne et al. (2009)	miR-21, miR-181b, miR-345, miR-1290, miR-1, miR-17-5p, miR-106b, miR-133a, miR-133b, miR-146a, miR-184, miR-196a, miR-206, miR-518b, miR-520g, miR-649	FFPE tissue	TLDA (Applied Biosystems)	qRT-PCR TaqMan MicroRNA Assay (Applied Biosystems)	RNU44
Yang et al. (2013)	Tissue: miR-10b-5p, miR-99a-5p, miR-99b-5p, miR-145-5p, miR-100-5p, miR-125b-5p, miR-181b, miR-181c, miR-197-3p, miR-331-3p, miR-15a-5p, miR-708, miR-150-5p, miR-30e-3p, miR-30a-3p, miR-21, let-7a-5p, miR-335-5p, miR-144*, miR-25-3p, miR-19a-3p, miR-660-5p, miR-140-5p, miR-590-5p, miR-9. Saliva: miR-10b, miR-145, miR-99b, miR-708, miR-181c, miR-30e, miR-660 and miR-197	Fresh frozen tissue, Saliva	Microarray microRNA Global expression analysis TaqMan low density array (TLDA)	qRT-PCR	RNU6
Santhi et al. (2013)	miR-125a, miR-184, miR-16, miR-96	Fresh frozen tissue	OrCa-dB database	qRT-PCR TaqMan MicroRNA Assay (Applied Biosystems)	RNU-44
Brito et al. (2014)	miR-21, miR-345, miR-181b	Fresh frozen tissue, Blood	Literature	qRT-PCR TaqMan MicroRNA Assays (Applied Biosystems)	RNU-44
De Sarkar et al. (2014)	mir-1293, miR-31, miR-31*, miR-7, miR- 206, miR- 204, miR-133a, miR-1	Fresh frozen tissue	TLDA(Applied Biosystems)	qRT-PCR TaqMan MicroRNA Assays (Applied Biosystems)	RNU-44, RNU-48, U6/mmu6
Kao et al. (2015)	miR-21, miR-31, miR-146a, miR-184, miR-372, let7i	Saliva, Blood (Plasma)	Literature	qRT-PCR TaqMan MicroRNA Assays (Applied Biosystems)	U6 snRNA
Zahran et al. (2015)	miR-21, miR-184, miR-145	Saliva	Literature	qRT-PCR Real-time PCR was miScript SYBR green PCR kit (Qiagen)	SNORD68
Lu et al. (2015)	miR-196a, miR-196b	Blood (Plasma)	global profiling, Literature	qRT-PCR TaqMan microRNA Assay (Applied Biosystems)	NM
Harrandah et al. (2016)	miR-7, miR-21, miR-494, miR-375	FFPE tissue	Microarray analysis done previously	qRT-PCR TaqMan MicroRNA Assay (Applied Biosystems)	RNU 44
Chattopadhyay et al. (2016)	miR-7, miR-133a, miR-204, miR-206, miR-31, miR-31*, miR-1293	Fresh frozen tissue	NS	qRT-PCR TaqMan MicroRNA Assay (Applied Biosystems)	RNU-44
Hung et al. (2016)	miR-21, miR-31	Saliva, FFPE tissue	Literature	qRT-PCR TaqMan MicroRNA Assays (Applied Biosystems)	miR-16
Philipone et al. (2016)	miR-208b-3p, miR204-5p, miRNA-129-2-3p, miR-3065-5p	FFPE tissue	Training Cohort. Next Generation Sequencing (Illumina HiSeq 2500)	qRT-PCR TaqMan MicroRNA Assays (Applied Biosystems)	RNU48
Sun et al. (2016)	miR-9	Blood (Serum)	Literature	qRT-PCR	RNU6B
Prasad et al. (2017)	miR-24, miR-26b, miR-155, miR-21, miR-31, miR-127, miR-197, miR-210, miR-19b, miR-205	FFPE tissue	Literature review	qRT-PCR TaqMan hydrolysis probes (Life Technologies)	miR-19b, miR-31, miR-205, miR-210
Chang et al. (2018)	miR-222-3p, miR-423-5p, miR-150-5p	Blood (Plasma)	Small RNA-sequencing, qRT-PCR.	qRT-PCR miScript SYBR Green PCR (Qiagen)	miR-130b-3p, miR-221-3p
Chen et al. (2018)	miR-129-5p, miR-296-5p and miR-450b-5p	FFPE tissue	gene expression omnibus (GEO) datasets	Real-Time PCR detection system (Bio-Rad Laboratories)	U6
Wang et al. (2018)	miR-31	Fresh frozen tissue, Blood (serum)	Literature	qRT-PCR TaqMan MicroRNA Assays (Applied Biosystems)	RNU6B

NS: Not Specified.

**Table 4 biomolecules-11-01879-t004:** Most-investigated miRNAs in relation to OLK/OED and OSCC and key findings.

miRs	No. of Included Studies That Have Investigated, Direction of Dysregulation Reported			Fold Change OLK/OED vs. N	Fold Change OSCC vs. OLK/OED	Fold Change OSCC vs. N	Fold Change Progressive OLK/OED vs. Non Progressive OLK/OED
miR-21	8 **	(Cervigne et al., 2009) 🡹		🡹 2.846 in dysplasia (*p* < 0.01)	NS	🡹 3.988 in OSCC (*p* < 0.01)	🡹 in progressive
		(Yang et al., 2013) 🡹		NS	NS	NS	🡹 5.20 in progressive (*p* < 0.049)
		(Brito et al., 2014) 🡹		🡹 in OLK (*p* = 0.01)	🡹 in OSCC (*p* = 0.02)	🡹 in OSCC (*p* = 0.01)	No statistical difference regarding miR expression was observed among the OLK group according to the severity of dysplasia.
		(Kao et al., 2015) 🡹		🡹 in dysplasia	🡹 in OSCC	🡹 IN OSCC	NS
		(Zahran et al., 2015) 🡹		With OED: 🡹 in OPMD Without OED: 🡹 in OPMD (*p* < 0.001)	With OED: 🡹 in OSCC Without OED: 🡹 in OSCC	🡹 in OSCC (*p* < 0.001)	NS
		(Harrandah et al., 2016) 🡹		NS	🡹 in OSCC (*p* = 0.0069)	🡹 in OSCC	NSD
		(Hung et al., 2016)	🡹 Saliva	🡹 in OPMD (*p* <0.003)	NS	NS	
			🡹Tissue	NS	NSD	NS	OPMD lesion with progression: Nss OPMD lesion without progression: miR-21 staining pixel intensity in the epithelium of OPMD = 35% median
		(Prasad et al., 2017) 🡹		CNRQ ratios: OED/HNE: 🡹 3.17 in dysplasia (*p* < 0.05)	🡹 in OSCC	CNRQ ratios: OSCC/HNE: 🡹 6.25 in OSCC (*p* < 0.05)	NS
miR 31	6 **	(De Sarkar et al., 2014) 🡹		🡹 4.55 in OLK	NS	🡹 5.37 in OSCC (*p* = 0.0006)	NS
		(Kao et al., 2015) 🡹		🡹 in dysplasia	🡹in OSCC	🡹 in OSCC	NS
		(Chattopadhyay et al., 2016) 🡹		🡹 68 in OLK (*p* < 0.05)	NS	Paired samples: 🡹 5 in OSCC (*p* < 0.05) unpaired samples: 🡹 30 in OSCC (*p* < 0.05)	🡹 (data not shown)
		(Hung et al., 2016)	Saliva: 🡹	Saliva: 🡹 in OPMD (*p* < 0.001)	🡹 (*p* = 0.01)	🡹 in OSCCC	NS
			Tissue: 🡹	Tissue: 🡹	NS	NS	In progressive OPMD lesions: 🡹 staining (*p* = 0.01) In non-progressive OPMD lesions: mir-31 staining pixel intensity in epithelium =57% median
		(Wang et al., 2018) 🡹		🡹 in OED (*p* < 0.01)	🡹 in HNSCC (*p* < 0.01)	Tissue: 🡹 in HNSCC (*p* < 0.01) Blood: 🡹 in HNSCC (*p* < 0.01)	🡹 in progressive
		(Prasad et al., 2017) unchanged		Unchanged (*p* > 0.05)	NS	Unchanged (*p* > 0.05)	NS
miR 184	4	(Cervigne et al., 2009) 🡹		🡹 1.86 in dysplasia (*p* = 0.0094)	NS	🡹 2.388 in OSCC (*p* < 0.01)	🡹 in progressive
		(Santhi et al., 2013) 🡻		🡻 0.14 in OLK (*p* < 0.0001)	🡻 in OSCC	🡻 0.1 in OSCC (*p* < 0.0001)	NS
		(Kao et al., 2015) 🡹		🡹 in dysplasia	NS	🡹 in OSCC	NS
		(Zahran et al., 2015) 🡹		With OED: 🡹 in OPMD (*p* < 0.001) Without OED: 🡹 in OPMD (*p* < 0.001)	With OED: 🡹 3 in OSCC (*p* < 0.001) Without OED: 🡹 in OSCC (*p* < 0.001)	🡹 in OSCC (*p* < 0.001)	NS
miR 181b	3	(Cervigne et al., 2009) 🡹		🡹 2.19 in dysplasia (*p* = 0.00032)	NS	🡹 3.539 in OSCC (*p* < 0.01)	🡹 in progressive
		(Yang et al., 2013) 🡻		NS	NS	NS	🡻 14.05 in progressive (*p* < 0.016)
		(Brito et al., 2014) 🡹		NS	🡹 in OSCC (*p* = 0.02)	🡹 in OSCC (*p* = 0.05)	No statistical difference regarding miR expression was observed among the OL group according to the severity of dysplasia.
miR 204	3	(Chang et al., 2008) 🡻		NS	NS	🡻 expression in 78% of OSCC compared to paired non-cancerous mucosa in same patient	NS
		(De Sarkar et al., 2014) 🡻		🡻 1.99 in OLK (Nss)	NS	🡻 27.02 IN OSCC (*p* = 0.0004)	NS
		(Chattopadhyay et al., 2016) 🡹🡻		🡹 4 in OLK(Nss)	NS	Paired samples: 🡻27 in OSCC (*p* < 0.05) Unpaired samples 🡻23 in OSCC (*p* < 0.05)	NS
miR 7	3 **	(De Sarkar et al., 2014) 🡹		1.18 (Nss)	NS	🡹 3.89 in OSCC (*p* = 0.0004)	
		(Harrandah et al., 2016) 🡹		NS	NS	🡹 in OSCC	NSD
		(Chattopadhyay et al., 2016) 🡹		🡹 6 in OLK (*p* < 0.05)	NS	Paired samples: 🡹 4 (*p* < 0.05) Unpaired samples: 🡹 2 (*p* < 0.05)	NS
miR 133a	3	(Cervigne et al., 2009) 🡻		🡻 0.016 in dysplasia Nss (*p* = 1)	NS	🡻 0.259 in OSCC Nss (*p* = 0.99)	🡻 in progressive
		(De Sarkar et al., 2014) 🡻		1.46 (Nss)	NS	🡻 97.14 in OSCC (*p* = 0.00005)	NS
		(Chattopadhyay et al., 2016) 🡹🡻		🡹 20 in OLK (*p* < 0.05)	NS	Paired samples: 🡻 97 in OSCC (*p* < 0.05) Unpaired samples: 🡻 209 in OSCC (*p* < 0.05)	NS
miR 206	3	(Cervigne et al., 2009) 🡹🡻		🡻 0.074 in dysplasia Nss (*p* = 0.99)	NS	🡹 4.286 in OSCC (*p* < 0.005)	🡻 in progressive dysplasia and subsequently 🡹 in OSCC
		(De Sarkar et al., 2014) 🡻		1.34 (Nss)	NS	🡻 31.54 in OSCC (*p* = 0.0001)	NS
		(Chattopadhyay et al., 2016) 🡹🡻		🡹 22 in OLK (*p* < 0.05)	NS	Paired samples: 🡻 32 in OSCC (*p* < 0.05) Unpaired samples:🡻 126 in OSCC (*p* < 0.05)	NS
miR 31*	2 **	(De Sarkar et al., 2014) 🡹		🡹 4.75 in OLK Nss	NS	🡹 6.73 in OSCC (*p* = 0.00005)	NS
		(Chattopadhyay et al., 2016) 🡹		🡹 65 in OLK (*p* < 0.05)	NS	Paired Samples: 🡹 7 in OSCC (*p* < 0.05) Unpaired samples: 🡹 41 in OSCC (*p* < 0.05)	🡹 in progressive
miR 1293	2 **	(De Sarkar et al., 2014) 🡹		1.18 (Nss)	NS	🡹 4.99 in OSCC (*p* = 0.000028)	NS
		(Chattopadhyay et al., 2016) 🡹		🡹 33 in OLK (*p* < 0.05)	NS	Paired samples:🡹 5 (*p* < 0.05) Unpaired samples:🡹 5 (*p* < 0.05)	NS
miR 1	2 **	(Cervigne et al., 2009) 🡻		🡻 0.008 in dysplasia (*p* = 1) (Nss)	NS	🡻 0.123 in OSCC (*p* = 0.99) Nss	🡻 in progressive
		(De Sarkar et al., 2014) 🡻		NS	NS	🡻 in OSCC Nss (*p* = 0.0009)	NS
miR 196a	2	(Cervigne et al., 2009) 🡹🡻		🡻 0.519 in dysplasia Nss (*p* = 0.16)	NS	🡹 4.185 in OSCC (*p* < 0.005)	🡻 in progressive and subsequently 🡹 in OSCC
		(Lu et al., 2015) 🡹		🡹 5.9 in OPMD (*p* < 0.001) OR 19.1 (*p* < 0.0001)	NS	🡹 9.3 in OSCC (*p* < 0.01) OR 51 (*p* < 0.0001)	NS
miR 9	2 **	(Yang et al., 2013) 🡻		NS	NS	NS	🡻 8.92 in progressive (*p* < 0.039)
		(Sun et al., 2016) 🡻		🡻 in OLK (*p* < 0.01)	🡻 in OSCC (*p* < 0.05)	🡻 IN OSCC (*p* < 0.01)	NS
miR 146a	2 **	(Cervigne et al., 2009) 🡹		🡹 5.031 in dysplasia (*p* = 0.00019)	NS	🡹 7.547 in OSCC (*p* < 0.01)	🡹 in progressive
		(Kao et al., 2015) 🡹		🡹 in dysplasia	🡹 in OSCC	🡹 in OSCC	NS
miR 345	2 **	(Cervigne et al., 2009) 🡹		🡹 2.073 in dysplasia (*p* < 0.01)	NS	🡹 3.528 in OSCC (*p* < 0.01)	🡹 in progressive
		(Brito et al., 2014) 🡹		NS	🡹 in OSCC (*p* = 0.0002)	🡹 in OSCC (*p* = 0.005)	No statistical difference regarding miR expression was observed among the OL group according to the severity of dysplasia.

** miRNA dysregulation consistent across studies. OSCC: Oral Squamous Cell Carcinoma; N: normal mucosa; OLK: Oral Leukoplakia; OED: Oral Epithelial Dysplasia; NS-not specified; NSS-not statistically significant; OPMD: Oral Potentially Malignant Disorders; NSD: No significant difference.

**Table 5 biomolecules-11-01879-t005:** MiRNAs predicting malignant transformation in oral leukoplakia and oral epithelial dysplasia.

Author, Year	MiRNA Investigated	Sensitivity (%)	Specificity (%)	ROC	AUC
Lu et al., 2015	miR-196a	(Potential malignancy detection OPL + 0SCC) = 64.2	(Potential malignancy detection OPL + 0SCC) = 96.2	Yes	Combining OPL and OSCC as diseases groups to see potential malignancy group Vs Normal = 0.848
	miR-196b	(Potential malignancy detection OPL + 0SCC) = 93.4	(Potential malignancy detection OPL + 0SCC) = 81.1	Yes	Combining OPL and OSCC as diseases groups to see potential malignancy group Vs Normal = 0.947
	miR-196a + miR-196b (together)	(Potential malignancy detection OPL + 0SCC) = 90.6	(Potential malignancy detection OPL + 0SCC) = 84.9	Yes	Combining OPL and OSCC as diseases groups to see potential malignancy group Vs Normal = 0.950
Harrandah et al., 2016	miR-21	Progressive vs non-progressive OPMD = 51.61	Progressive vs non-progressive OPMD = 83.33	Yes (to differentiate progressive from nonprogressive premalignant lesions)	0.651 (*p* = 0.18)
	miR-375	Progressive vs non-progressive OPMD = 90.32	Progressive vs non-progressive OPMD = 100	Yes (to differentiate progressive from nonprogressive premalignant lesions)	0.957 (*p* < 0.0001)
	miR-375/miR-21	Progressive vs non-progressive OPMD = 75.76	Progressive vs non-progressive OPMD = 100	Yes (to differentiate progressive from nonprogressive premalignant lesions)	0.925 (*p* < 0.0001)
Hung et al., 2016	miR-31	87.51	73.73%	Yes	0.81
	miR-21	NSS	NSS	Yes	0.56
Philipone et al., 2016	miR-208b-3p, miR-3065-5p, miR-129-2-3p, miR-204-5p	76.9	73.7	Yes	0.792
Chen et al., 2018	miR-129-5p, miR-296-5p and miR-450b-5p	Only graph shown, data not shown	NS	Yes	miR-129-5p: 0.73 miR296-5p: 0.759 miR-450b-5p: 0.721 Combination of these three miRNAs signatures: 0.872

ROC: Receiver Operating Characteristic; AUC: Area Under the Curve; OPL: Oral Premalignant Lesions; OSCC: Oral Squamous Cell Carcinoma; OPMD: Oral Potentially Malignant disorders.

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
