# Peer review of "A Systematic Review of MicroRNA Signatures Associated with the Progression of Leukoplakia with and without Epithelial Dysplasia"

_biomolecules, 2021, doi:10.3390/biom11121879_

Round 1

Reviewer 1 Report

I would suggest minor changes of the “Results and Discussion” section, that needs to follow a more concise and precise order of ideas related to the Table 2 and Table 3. Additionally, this can better guide the reader toward the area of “Future directions and Conclusions”.

Since there has been evidence that miR-375 could be significant in differentiating between progressive and non-progressive premalignant lesions, I suggest that the authors of the manuscript provide more detailed information and describe the most relevant biomarker for this certain miRNA type.

Reviewer 2 Report

The review is well-written and thoroughly reviews the field of using differentially expressed miRNAs as markers for oral cancer diseases. 

The overall background of the relevant diseases as well as the significance of miRNAs as biomarkers are described in detail.

The methods for the thorough literature research are exceptionally well and comprehensibly described. The resulting publications of the literature search are extensively analyzed and the described miRNAs are listed ordered according to publication. This gives a very orderly overview of the miRNAs discussed including the used methods. The quality of the studies is critically discussed with respect to differentiations of the several oral diseases.

The benefits of using single or a combination of multiple miRNAs as predictors is discussed. 

Author Response

Thank you very much for the feedback. 

Reviewer 3 Report

This is the systematic review about miRNA expression in OPMDs. The aim and methodology are clear, and results seem to be solid. This review will provide meaningful knowledge about miRNAs profile during OSCC carcinogenesis to journal readers.

Author Response

Thank you very much for the feedback.